# Single/Dual/Triple Broadband Metasurface Based Polarisation Converter with High Angular Stability for Terahertz Applications

**DOI:** 10.3390/mi13091547

**Published:** 2022-09-18

**Authors:** Shyam Sundar Pati, Swaroop Sahoo

**Affiliations:** Indian Institute of Technology Palakkad, Palakkad 678623, India

**Keywords:** metamaterial, terahertz, polarisation converter, bandwidth, high impedance surface, angular stability, ultra-wideband

## Abstract

This paper presents design and characterisation of a new compact metasurface based linear polarisation converter for terahertz applications. The metasurface unit cell with periodicity of 0.292$\lambda_{0}$ consists of an asymmetrically oriented planar double semicircular goblet-shaped resonators. It is printed on a polydimethylsiloxane (PDMS) dielectric substrate backed by a gold layer that acts as a ground plane. This metasurface structure exhibits a broadband cross-polarisation conversion in the frequency range of 0.72–0.99 THz with a polarisation conversion ratio (PCR) > 95% and angular stability > ${\mathrm{40}}^{\circ }$ for both TE and TM modes. However, the PCR for the single band is >99% at resonant frequencies of 0.755 and 0.94 THz, while the optimised design shows 100% PCR over a BW of 95 GHz. Furthermore, slight modification and optimisation of the broadband design results in quad-ring and slotted DSGRs that produce dual and triple broadband polarisation conversion, respectively. The quad-ring DSGR performs polarisation conversion for frequency range of 0.70–1.08 and 1.61–1.76 THz while the slotted DSGR shows the triple broadband cross-conversion for frequency range of 0.67–0.85, 1.04–1.11, and 1.62–1.76 THz with PCR > 95%. This design is simple, easy to modify to implement single and multi broadband polarisation conversion with high PCR at terahertz regime. In addition to that, it is easy to fabricate and integrate with other components like multiple-input multiple-output terahertz antennas for mutual coupling reduction.

## 1. Introduction

Terahertz (THz) frequency range is the intermediate frequency band (0.1–10 THz) between microwave and infrared frequencies in the electromagnetic (EM) spectrum whose wavelength is in the range of 0.03–3 mm [1]. Recent studies in this THz range have shown the various potential applications in different domains like enhancing the rate of signal and data transmission in wireless communication, designing high resolution sensor [2], time domain spectroscopy for determining the real and imaginary values of the dielectric constant of a material [3] and cancer detection [4]. Therefore, different EM devices are being designed to meet the above objectives for altering phase, amplitude, and polarisation of THz signals. Among the above mentioned signal properties, polarisation states of THz waves can be easily manipulated using polarisation converters (PC) [5]. Here, a novel PC (in the terahertz regime) has been designed and simulated using metasurface for achieving single, dual and triple broadband orthogonal polarisation conversion.

PCs have various applications for antenna gain enhancement, multiple-input multiple-output (MIMO) antenna isolation, reduction of radar cross section [6], fiber optics, liquid crystal display, nanophotonic sensing systems, microwave and terahertz communications [7,8]. PCs can be of reflective/transmissive types that can convert linear to linear, linear to circular, right (left) circular to left (right) circular polarisation [9]. Traditional PCs are based on the Faraday effect, optical grating and anisotropic and birefringence crystal effects; but they have high profile, limited broadband response [10] and significant reflection losses [11]. Therefore, PCs are being designed in the terahertz regime, using metamaterials (MMs) that are compact in nature and have broadband polarisation conversion. Metamaterials are artificially engineered materials that have exotic properties like negative electric permittivity and magnetic permeability. It can easily alter the polarisation states of EM waves by manipulating surface waves by portable planar structures called metasurfaces (2D equivalent of 3D MM) [12,13,14,15,16,17,18,19]. Metamaterials can be used for designing highly sensitive refractive sensors based on strong magnetic resonance [20]. These materials can be also used for designing metal-dielectric-metal waveguide based plasmonic guiding structures due to ease of fabrication and zero bending losses [21,22]. Further studies on MMs have proven that spin-orbit interaction of light, cloaking and holograms can be achieved by employing MMs for full-stokes polarisation control. Recently, full-stokes polarisation perfect absorbers were designed using diatomic plasmonic metasurface [23].

Existing techniques for achieving cross-polarisation conversion include crystal based birefringent wave plates [24] and planar unit cells of different shapes and sizes like metallic grating layers coupled to subwavelength Si grating [25], double gold layer slot structure [26], cut wire resonators [27], symmetric and asymmetric crosses on dielectric substrate [28], double split ring resonator [29], Fabry-perot cavity resonator [30] and T-shaped resonator [31].

The terahertz converters realized using different methods in [24,25,26,27,28,29,30,31] have good linear cross-conversion but have certain limitations. The crystal based THz polarisation converter in [24] resulted in a bulky design. The converters in [25,26] rotate the incident electric field polarisation by $90^{\circ }$ in transmissive mode; but have significant losses. The angular stability of the above converters is not studied well. In [27], the PCR is up to 80% both for transmissive and reflective modes. But, the metallic slot arrays created for getting polarisation conversion at transmissive mode hinder the performance of the converter for reflective mode in the terahertz regime. In addition, transmissive type PCs are very difficult to fabricate and are lossy due to which multi-layer stacking is needed for loss minimization. Thus, reflective type PCs have been used a lot due to their ease of fabrication and design for cross-conversion. The converters realized in [28,29,30,31] are reflective in nature. But all the above realisations are broadband in nature and are not suitable for multifunctional applications where the system operates at multiple bands. This is significant in multiple input and multiple output terahertz antennas operating at different frequencies. One of the major problems in the case of MIMO antennas is the mutual coupling between elements that impact the performance. Recent studies says that metasurface integrated MIMO antennas with multiband polarisation converters can perform the coupling reduction to a greater extent. Thus, multiband polarization converters are very important for terahertz applications.

Considering the above limitations of broadband polarisation converters, a new ultra-thin and ultra-wideband reflective type polarisation converter is designed and simulated. This PC operates in THz band with a PCR value greater than 95% over the defined frequency band. A slight optimization of the design results in 100% ideal cross-polarisation conversion over a wide frequency band with angular stability of up to $45^{\circ }$ both for transverse electric (TE) mode and transverse magnetic (TM) mode. This single broadband PC can be easily optimized for dual and triple broadband operation. The above design is very simple but performs well for different single and multiband terahertz polarisation conversion applications. In this article, Section 2 presents the step-by-step design process of the metasurface unit cell based on co-pol and cross-pol reflection coefficients. This is followed by performance analysis and simulation results of the broadband PC. For a better understanding of the polarisation conversion mechanism, U-V decomposition and impedance imbalance for in-phase reflection have been also discussed. Along with that oblique angle stability and fabrication tolerance of the proposed PC have been studied. Section 3 shows the designs for multiband PC along with simulation and analysis results. The paper is concluded in Section 4 with a state-of-the-art performance comparison table of the designed PCs with previously reported structures.

## 2. Design Principle of Polarisation Converter

The operation of a reflective type polarisation converter can be explained on the basis of a electromagnetic absorber. The absorptivity of an isotropic reflective metasurface-based EM absorber is defined as $A=1-S_{11}^{2}-S_{21}^{2}$, where $S_{11}$ and $S_{21}$ are reflection and transmission coefficients, respectively [32]. The absorptivity can be modified as $A=1-S_{1x}^{2}-S_{1y}^{2}$ (where $S_{21}=0$) when the incident EM waves are x- and y-polarised and the metasurface is backed by a bottom metallic ground plane. The absorption of EM wave is ideal when co-pol ($S_{1x}$ or $S_{1y}$) and cross-pol ($S_{1y}$ or $S_{1x}$) reflection coefficients are as low as possible. In the ideal case, the cross-polarised reflection coefficient will exceed the co-polarised component due to anisotropic effect if the symmetry of the metasurface is broken. The above statement signifies that a y- or x-polarized EM wave can be converted to its orthogonal counterpart. Further, the analysis of controlling cross-pol and co-pol reflection coefficients of the anisotropic geometry can also be extended to scattering matrix, impedance matching and material properties ($\left(\mu_{r}\right)$ and $\left(\epsilon_{r}\right)$). This is because impedance matching and material properties play a key role in absorption and polarisation conversion and have been analyzed extensively in later sections. The normalised effective impedance, permittivity $\left(\epsilon_{r}\right)$, and permeability $\left(\mu_{r}\right)$ of the anisotropic metasurface can be calculated from the reflection and transmission coefficients i.e., $S_{11}$ and $S_{21}$ as per the Equation (Equation 1) [32,33]:(1)$$Z_{eff}=\pm \sqrt{\frac{{\left(1+S_{11}\right)}^{2}-S_{21}^{2}}{{\left(1-S_{11}\right)}^{2}-S_{21}^{2}}}=\frac{1+S_{11}}{1-S_{11}}$$
(2)$$\epsilon_{eff}=\frac{n}{Z_{eff}}$$
(3)$$\mu_{eff}=n\times Z_{eff}$$
where *n* is known as the refractive index of the unit cell.

For determining the co-pol and cross-pol reflection coefficients, the EM wave with a specified linear polarisation is incident on the metasurface. The final reflected wave obtained through multiple reflections between the top metasurface and bottom ground plane contains both co- and cross-polarised EM waves. The incident and reflected EM waves are related to each other by the Equation (Equation 4) [34]:(4)$$\left(\begin{array}{c} E_{rx}\\ E_{ry} \end{array}\right)=\left(\begin{array}{cc} r_{xx} & r_{xy}\\ r_{yx} & r_{yy} \end{array}\right)\left(\begin{array}{c} E_{ix}\\ E_{iy} \end{array}\right)=R_{jk}\left(\begin{array}{c} E_{ix}\\ E_{iy} \end{array}\right)$$
where ${\left[R_{jk}\right]}_{j,k=x,y}$ stands for the linearly polarised reflection coefficient matrix and $r_{jj}\left(j=k\right)$ and $r_{jk}\left(j\neq k\right)$ are the co-pol and cross-pol reflection coefficients, respectively. The operation principles of a PC are mentioned in Appendix A.

Based on the above discussion, the structure for the ultra-thin and ultra-wideband reflective type angular stable linear polarisation converter at terahertz regime has been proposed. The proposed unit cell of the single broadband converter consists of a double semi-circular goblet-shaped resonator (DSGR) placed at an angle of 45° with respect to the axis of the cell. The geometrical parameters of the proposed cross converter have been optimized for obtaining maximum bandwidth and are given in Table 1. The overall geometry of the proposed PC consists of three layers: a top metasurface consisting of DSGR unit cells made up of gold with electrical conductivity $\sigma =4.10\times 10^{7}$ S/m, intermediate PDMS dielectric substrate of length 116 µm and complex dielectric constant $\epsilon_{r}=2.35+j0.06$ and bottom metallic ground plane of gold, as depicted in Figure 1. The gold thickness both for metasurface and the bottom ground plane is 200 nm while the dielectric thickness is 33 μm.

As mentioned above, the symmetry of the structure should be broken to create the anisotropy which in turn will result in a higher cross-polarisation component than the co-polarisation component. In order to achieve this, five consecutive design steps have been followed with their corresponding co-pol $r_{yy}$ and cross-pol $r_{xy}$ reflection coefficients shown in Figure 2. The simulated values of $r_{yy}$ and $r_{xy}$ corresponding to five steps have been given in Table 2. For this analysis, a y-polarised EM wave in the frequency range of 0.2 to 1.2 THz is used. It is obvious from the results that adding two semi-circular rings of width 5 μm to individual strips in mirror symmetric configuration results in efficient polarisation conversion with $r_{xy}$ > −2 dB and $r_{yy}$ < −16 dB throughout the operating BW of 0.72–0.99 THz. The two co-pol resonance dips are observed at 0.76 and 0.945 THz and are found to be <−20 dB.

### 2.1. Performance Analysis of the Single Broadband Polarisation Conversion Metasurface

The design and simulation of the proposed wide-broadband reflective type linear polarisation converter has been done using ANSYS High Frequency Structure Simulator (HFSS) of ver. 2020. The designed unit cell is shown in Figure 1a,b and is periodically repeated to form the 2D metasurface PC shown in Figure 3a. The performance analysis of the proposed converters is carried out by imposing periodic Master-Slave boundary conditions along x- and y-directions as exhibited in Figure 1c. When the EM wave is incident on the metasurface, it produces both co- and cross-polarised reflected waves with reflection coefficients $r_{yy}$ and $r_{xy}$. The co-pol and cross-pol reflection coefficients for a y-polarised incident wave are defined as $r_{yy}=20\log\left(\frac{\vec{E_{yr}}}{\vec{E_{yi}}}\right)$ and $r_{xy}=20\log\left(\frac{\vec{E_{xr}}}{\vec{E_{yi}}}\right)$, respectively. In this analysis, the incident and reflected waves contain both TE and TM modes where most of the analyses have been performed using y-polarised TE mode. For normal incidence, a y-polarised TE or TM mode is considered as the diagonal symmetry of the unit cell structure giving similar response for each mode. However, to verify the angular stability of the structure for different oblique angles of incidence, both TE and TM modes are taken into consideration.

As already discussed, the basic performance analysis of a PC is determined based on co-pol and cross-pol reflection coefficient values where the co-pol reflection coefficient should be as minimum as possible in comparison to the cross-pol reflection coefficient. Here the threshold value for the reflection coefficients have been taken as $r_{yy}$ < −10 dB and $r_{xy}$ > −3 dB. In addition to that, the polarisation conversion ability of the metasurface is characterized by the polarisation conversion ratio (PCR) and mathematically it is defined as the ratio of cross-pol reflected power to the sum of cross-pol and co-pol reflected powers. It can be defined in terms of reflection coefficients as shown in Equation (Equation 5) [17]
(5)$$PCR=\frac{r_{xy}^{2}}{r_{xy}^{2}+r_{yy}^{2}}$$

The PCR for proposed converter with strip length $l_{1}=117$
μm (see Figure 1a) has been shown in Figure 3b. The PCR is greater than 95% for this device for the frequency range of 0.72–0.99 THz (270 GHz BW). Specifically, this converter shows two dominant resonances at 0.755 and 0.94 THz with PCR > 99% while the PCR corresponding to the central dip is 97%. Further optimization of the strip-length of the goblet shaped resonators has been performed through parametric variation to achieve an ideal 100% polarisation conversion, but at the expense of bandwidth. This optimization results in strip length $l_{1}=110$
μm that gives a PCR of 100% over the frequency range of 0.85–0.945 THz (i.e., 95 GHz BW) as shown in Figure 4a. For such wide-broadband ideal cross-polarisation conversion, $r_{yy}$ should be much lower than $r_{xy}$ as obvious from Figure 4b. The optimized wide-broadband converter with PCR of 100% has $r_{yy}$ < −25 dB and $r_{xy}$ > −2 dB throughout the operating BW.

### 2.2. U-V Decomposition Analysis

A mathematical discussion has been presented here for a better understanding of the anisotropic medium characteristics of the proposed converter and the conversion mechanism. This analysis has been performed by defining another rectangular co-ordinate system, u−v that makes an angle of $45^{\circ }$ with respect to the original x−y coordinate axes as shown in Figure 5. Here the analysis has been restricted only to y-polarised EM wave but its analogous x-polarised wave will also give the same results. The general expressions for x- and y-polarised incident EM waves are given as $\vec{E_{xi}}=\left|E_{xi}\right|\vec{i}$ and $\vec{E_{yi}}=\left|E_{yi}\right|\vec{j}$ where $\vec{i}$ and $\vec{j}$ are the unit vectors along x- and y-axis. Due to the diagonal orientation of the DSGR, the y-polarised EM wave incident on the metasurface will be decomposed into components along u- and v- axis resulting in $\vec{E_{y_{i}}}=|\vec{E_{iu}}|e^{j\varphi }\hat{e_{1}}+\left|\vec{E_{iv}}\right|e^{j\varphi }\hat{e_{2}}$ where $\vec{e_{1}}$ and $\vec{e_{2}}$ are the unit vectors along u- and v-axis and φ is the phase with respect to the original co-ordinate system. The EM wave transmitted through the metasurface will undergo multiple reflections between the top metasurface and bottom ground plane. The resultant reflected after constructive interference produces an x-polarised EM wave that can be written as $\vec{E_{xr}}=|\vec{E_{ru}}|e^{j\varphi_{1}}\hat{e_{1}}+\left|\vec{E_{rv}}\right|e^{j\varphi_{2}}\hat{e_{2}}$ where $\varphi_{1}$ and $\varphi_{2}$ are the phases of reflection coefficients $r_{u}$ and $r_{v}$ along u- and v-axis. As u- and v-axis are rotated about an angle of $45^{\circ }$ relative to x- and y-axis, the most general form of $\vec{E_{xr}}$ and $\vec{E_{yi}}$ can be expressed as Equations (Equation 6)–(Equation 8):(6)$$\vec{E_{y_{i}}}=|\vec{E_{iu}}|e^{jkz}\hat{e_{1}}+|\vec{E_{iv}}|e^{jkz}\hat{e_{2}}=\left|\vec{E}\right|cos\left(45^{\circ }\right)e^{jkz}\left({\hat{e}}_{1}+{\hat{e}}_{2}\right)$$
(7)$$\vec{E_{xr}}=|E_{ru}|\hat{e_{1}}+|E_{rv}|\hat{e_{2}}=r_{u}|E_{iu}|\hat{e_{1}}+r_{v}\left|E_{iv}\right|\hat{e_{2}}$$
(8)$$\Rightarrow |E_{y_{i}}|\left(cos\left(45^{\circ }\right)e^{-jkz+\varphi_{1}}\hat{e_{1}}+cos\left(45^{\circ }\right)e^{-jkz+\varphi_{2}}\hat{e_{2}}\right)$$

The co-polarised reflection coefficients $r_{u}$ and $r_{v}$ are independent of each other due to the anisotropic geometry of the unit cell and are connected to u- and v-axis just like co-pol and cross-pol reflection coefficients ($r_{xx}$ and $r_{xy}$) as: $|r_{u}|$=$|\frac{\vec{E_{ru}}}{\vec{E_{iu}}}|$ and $|r_{v}|$=$|\frac{\vec{E_{rv}}}{\vec{E_{iv}}}|$. Cross-polarisation conversion for a reflective type linear cross converter will happen if the orthogonal co-polarised coefficients $r_{u}$ and $r_{v}$ are equal in magnitude (ideally unity) and the phase difference $\left(\Delta \varphi =\varphi_{1}-\varphi_{2}\right)$ between them is $180^{\circ }$ i.e., $r_{u}\approx r_{v}$ and $\Delta \varphi =\pm 180^{\circ }$. Hence, a y-polarised incident electric field $\vec{E_{y}}$ will give rise to an x-polarised reflected electric field $\vec{E_{x}}$.

To verify the above conditions for polarisation conversion, the magnitudes of the co-polarised reflectance $r_{u}$ and $r_{v}$ for $l_{1}=117$
μm and $l_{1}=110$
μm along with the phase difference plot are shown in Figure 6a–c. It is observed that the magnitude of the orthogonal components are nearly equal to each other within the PCR defined BW. For $l_{1}=117$
μm, the PCR is taken over 95%, and the maximum PCR values (>99%) are obtained at the first and second resonances only i.e., at 0.755 and 0.94 THz, respectively. The phase plot in Figure 6c shows that $\Delta \varphi =\pm 180^{\circ }$ for those two resonance frequencies while for other frequencies in the PCR BW, it is $\Delta \varphi =\pm 180^{\circ }\pm \Delta x$ where Δx can be in the range of $\pm 15^{\circ }$. As explained before, the optimized strip length $l_{1}=110$
μm shows ideal 100% PCR from 0.85–0.945 THz, and the same has been depicted in phase plot i.e., phase difference between $r_{u}$ and $r_{v}$ exactly satisfies the condition for cross conversion i.e., $\Delta \varphi =180^{\circ }$ confirming the wide-broadband polarisation conversion.

### 2.3. Surface Current Distribution

The mechanism and working principle for the cross polarisation conversion in the case of a reflective type metasurface has been discussed in the beginning of Section 2. The interaction between EM wave and metasurface will result in electrically and magnetically polarised meta-atoms. These atoms in turn will induce electric and magnetic dipole moments i.e., $p_{e}$ and $m_{e}$ that will be coupled to both the E- and H-fields of EM wave. Due to the time harmonic nature of E- and H-fields, the permeability and permittivity will be frequency dependent ($\mu (\omega_{i})$ and $\epsilon (\omega_{i})$) and the effective surface impedance can be defined as $Z\left(\omega_{i}\right)=\frac{\mu (\omega_{i})}{\epsilon (\omega_{i})}$. The reflection coefficient can be written as Equation (Equation 9) [35]:(9)$$R\left(\omega_{i}\right)=\frac{Z\left(\omega_{i}\right)-Z_{0}}{Z\left(\omega_{i}\right)+Z_{0}}$$
(10)$$\Rightarrow R\left(\omega_{i}\right)=\frac{1-\frac{Z_{0}}{Z(\omega_{i})}}{1+\frac{Z_{0}}{Z(\omega_{i})}}$$
where $Z_{0}$ is defined as the free space impedance with value 377 Ω. At resonance frequency $\left(\omega_{ir}\right)$, $Z\left(\omega_{ir}\right)>>Z_{0}$ and the reflection coefficient $R(\omega_{i})$ will approach unity resulting in the structure behaving like a high impedance surface (HIS). In general, the incident and reflected waves for a normal reflective surface are out of phase from each other i.e., the phase difference between them is $180^{\circ }$. However, due to HIS the incident wave will be reflected with unity reflection coefficient without phase reversal. In addition, it is observed from the U-V decomposition analysis that the orthogonal components of the incident E/H fields are reflected by $0^{\circ }$ and $180^{\circ }$ phase resulting in $90^{\circ }$ rotation of plane of polarisation of the incident field. Thus, it can be claimed that the structure behaves as a high impedance surface for one component while acting as a common reflector for another component [34]. The general theory governing normal reflective surface exhibiting characteristics of a HIS can be analysed in the context of nature of resonance and Faraday’s law. As per the Faraday’s law, the time harmonic magnetic field (B) sandwiched between the top metasurface and bottom ground plane of the unit cell produces the surface current of opposite polarities on both sides. Due to the anti-parallel direction of surface current, the entire system exhibits a strong magnetic response. The permeability μ corresponding to the resonance frequency will be divergent in nature, leading to very high surface impedance and in-phase reflection.

The induced surface currents distribution on the top metasurface and bottom ground plane at resonant frequencies of 0.755 and 0.94 THz are shown in Figure 7 where the black arrow head denotes the net direction of the induced surface current. The surface current directions on the top and bottom ground plane of the unit cell are anti-parallel, confirming strong magnetic resonance at those resonance frequencies. The magnetic flux sandwiched between the top and bottom surface of the PC will be intensified in the substrate leading to increase of effective permeability which will increase the surface impedance of the unit cell. Hence, the normal metasurface unit cell starts behaving like an HIS and impedance imbalance occurs along the y-direction. This surface impedance gives rise to in-phase reflection with unity reflection coefficient at the resonant frequencies. As already mentioned, due to the phenomenon of impedance imbalance along the y-direction, the current is directed towards the x-direction at the resonant frequencies, resulting in an orthogonal polarised reflected wave in the x-direction.

### 2.4. Oblique Angle Stability

PCs are used for polarisation conversion at various angles of incidence and the change in angles of incidence will impact the PCR values as well as the bandwidth irrespective of the anisotropy of the metasurface. Therefore, the polarisation conversion ability of the proposed converter is analyzed for different oblique angles of incidence to determine its angular stability and efficiency. This analysis is performed assuming that the incident wave consists of transverse electric (TE) and transverse magnetic (TM) modes. Due to the change in oblique angles of incidence of the EM wave, the propagation phase (β) undergoes phase retardation and the values can be determined according to the Equation (Equation 11) [17]:(11)$$2\beta =2\sqrt{\frac{1+\epsilon_{r}}{2}}k_{0}\left(\frac{h}{cos\theta }\right)$$

Here, $k_{0}$ refers to the propagation constant of the EM wave in free space ($\frac{2\pi }{\lambda }$) and the factor of two in Equation (Equation 11) is due to the transmitted EM wave going through the metasurface and reflecting back after multiple reflections from the bottom ground plane.

Because of this reflection in the terahertz regime, there will be a significant change in the phase of EM wave resulting in destructive interference. This effect will reduce the operating bandwidth of the converter. The angular stability of the proposed single broadband PC (with strip lengths 110 and 117 μm) is simulated for various oblique angles from $0^{\circ }$ to $60^{\circ }$ as depicted in Figure 8 and Figure 9. Figure 8b,c show the PCR vs frequency plot for $l_{1}=117$
μm while using TE and TM modes. At $\theta =0^{\circ }$ and $15^{\circ }$, the PCR is >95% and the BW of the converter is 270 GHz (0.72–0.99 THz). However, the PCR and BW of the converter gradually decrease with increase in the oblique angle of incidence. It can be observed that the PCR corresponding to the first resonance (lower) frequency is least affected by the change in the angles of incidence for both TE and TM modes. On the other hand, there is a drastic change in the PCR values of the second resonant (higher) frequency with change in angles of incidence ($\theta >15^{\circ }$) as depicted in Figure 8b,c. At $\theta >40^{\circ }$, the BW reduces significantly because the significant reduction in PCR of higher resonant frequency. If PCR = 90% is taken as threshold, the structure is stable up to $\theta =30^{\circ }$ for both the TE and TM modes. It can also be observed that the bandwidth is atleast 70 GHz (0.73–0.80 THz) for PCR threshold of 90% for any angle of incidence considered. Furthermore, if the PCR thershold is taken to be >80%, the metasurface is observed to show angular stability of up to $40^{\circ }$ for TE mode and $45^{\circ }$ for TM mode. The optimized design with $l_{1}=110$
μm shows better angular stability than the former one with $l_{1}=117\;\mu$m and the results for TE and TM modes can be seen in Figure 9a,b. At $\theta =0^{\circ }$, the optimized PC has polarisation conversion BW of 95 GHz (0.85–0.945 THz) with an ideal PCR of 100%. With an increase in θ, the higher frequency band shifts to the left and the PCR value drops again. In general, the optimized design shows angular stability up to $35^{\circ }$ TE mode and $40^{\circ }$ TM mode with PCR > 90%.

### 2.5. Parametric Variation of Proposed Converter

The sensitivity of the proposed converter to variation of geometrical parameters such as the height of dielectric substrate, radii of the DSGR resonator, and strip length of the resonator is studied to demonstrate its fabrication tolerance. First the parametric variations is performed for different radii of the DSGR resonator and the impact on PCR is analysed in Figure 10a.

It is observed that decrease in radius from 32.5 to 20.5 μm results in shifting of the higher frequency band to right, thereby increasing the BW of the converter from 0.15 THz to 0.45 THz. However, the PCR value of the central dip gradually decreases and goes below 90% when the radii is decreased below 20.5 μm. The parametric variation of radii has a significant effect on the second resonance (higher) while there is minimal effect on the first resonance (lower). The parametric variation of the strip length $l_{1}$ (110 to 120 μm) of the resonator is performed (see Figure 10b). The increase in strip length beyond 117 μm shifts the lower resonant frequency to the left with minor changes to the higher one. This is due to the increase of the strip capacitance causing the frequency band to shift to the lower region. The increase in strip length increases the BW while the decrease in it enhances the PCR value (at the expense of the BW). At the optimized strip length of $l_{1}=110$
μm, the broadband cross-conversion occurs with a PCR value of exactly 100% as explained in the simulation results section.

The results for parametric variation of dielectric substrate height (h) has been exhibited in Figure 10c. It reveals that reduction of substrate height below 33 μm reduces the PCR value to 90% (at h=28
μm) while increase in height improves the PCR with narrower BW (less than that of substrate height of 33 μm). The authors have also performed parametric variation for the periodicity (L) and other physical dimensions which reveal that the structure is stable with slight change in PCR. Hence, the conclusion that is derived from the parametric variation is that the unit cell is highly stable to minor changes in the design parameters and, hence, it exhibits fabrication tolerance.

## 3. Multi Broadband Reflective Type Polarisation Converter

Multi-functionality applications in terahertz regime require multiband polarisation converters with high PCR. This can be achieved by slight modification of the broadband design to get dual and triple broadband converters. The dual-band PC is designed by adding a semi-circular ring of radius 8 μm to the existing broadband structure forming a semi-circular quad-ring unit cell. On the other hand the triple band PC is designed by introducing a slot of length $l_{s}=55$
μm and width $w_{s}=6.875$
μm in the metallic strips of the DSGR unit cell. This introduction of the slots results in more anisotropy to the structure and the layout of the multiband PCs are shown in Figure 11a,b. The optimized geometrical parameters of the multiband PCs are listed in Table 3 and Table 4.

### 3.1. Simulation Results of Dual Broadband PC

The performance of the dual band PC is determined based on the simulated PCR and reflection coefficient and the results are shown in Figure 12 while the PCR threshold for BW determination is taken to be >95%.

PCR values in Figure 12a show that first band has a BW of 380 GHz in the frequency range of 0.70–1.08 THz while the second band exhibits a BW of 150 GHz for the frequency range of 1.61–1.76 THz. The first band has two dominant resonances at 0.755 and 0.985 THz with PCR > 99.9%, while the resonance at 1.68 THz for the second band has a PCR of 98%. In addition to that, Figure 12b depicts that the co-pol reflection coefficient $r_{yy}$ (less than −20 dB) is much lower than that of $r_{xy}$ within the PCR defined BW. The two co-pol resonance dips in the first band have $r_{yy}$ < −22.5 and −30 dB while the second band resonance dip has $r_{yy}$ < −30 dB. For a y-polarised incident wave to be converted to an x-polarised reflected wave, the polarisation conversion conditions as mentioned in the above sections should be satisfied. For this purpose, the U-V decomposition analysis has been performed, (see Figure 12c,d). The dual broadband polarisation converter exhibits considerably better performance with higher PCR and angular stability as compared to a wide-broadband linear cross converter, as shown in the simulation results.

The dual-band performance of the proposed PC is also analyzed for different oblique angles of incidence to determine the angular stability. The performance has been analysed using TE and TM modes as incident waves and results can be observed from Figure 13a,c. The structure shows an angular stability of up to $30^{\circ }$ when PCR > 92% is the threshold for both the TE and TM modes. At the angle of incidence $30^{\circ }$, the BW of the first band is slightly improved while BW is shifted to the right (higher frequencies) in the case of the second band. However, at $30^{\circ }$ incident angle, the PC achieves an ideal 100% PCR in the spectrum (second broadband) centered at 1.685 THz for TE mode. An additional analysis has been performed as a special case for angular stability of the lower frequency band for TE and TM modes and the results are depicted in Figure 13b,d, respectively. The angular stability is determined for angles of incidence ranging from $0^{\circ }$ to $40^{\circ }$ and the simulation results show that the first band exhibits a trend similar to the single broadband DSGR unit cell. The achieved angular stability for TE and TM mode is up to $\theta =40^{\circ }$ when the PCR threshold is greater than 88%.

### 3.2. Simulation Results of Triple Broadband PC

The analysis of the reflective type triple band linear co- to cross-polarization converter is based on simulation results of PCR, reflection coefficients and phased difference shown in Figure 14. The PCR shown in Figure 14a determines that the PC achieves orthogonal polarization conversion with PCR > 95% for frequency ranges 0.67–0.85 (180 GHz BW), 1.04–1.11 (70 GHz BW) and 1.62–1.76 (140 GHz BW) THz. The centre frequencies of the first and third bands have ideal 100% PCR while the resonance at 1.07 THz (middle band) has a PCR of 98%. In addition to that, 100% cross-polarization conversion can be observed in the first band from 0.72 to 0.81 THz (i.e., BW of 90 GHz) and the third band from 1.67 to 1.70 THz (i.e., BW of 30 GHz).

To investigate the polarization conversion process, reflection coefficient analysis and the U-V decomposition have been carried out. Figure 14b shows the co-pol and cross-pol reflection coefficients (i.e., $r_{yy}$ and $r_{xy}$) where $r_{yy}$ < −15 dB and $r_{xy}\geq$−3 dB within the PCR defined BW. Three co-pol resonance dips are observed at 0.79, 1.07 and 1.68 THz. Another analysis i.e., the phase difference between $r_{u}$ and $r_{v}$ is shown in Figure 14c. The results show that $\Delta \varphi =180^{\circ }$ for the first and third bands (100% PCR regions), and $\Delta \varphi =180^{\circ }\pm \delta x$ for second and other PCR defined frequency spectra for PCR < 100%.

The triple band polarisation conversion efficiency has also been analyzed using different oblique angles of incidence. Figure 15 shows the PCR vs frequency plot for different oblique angles of incidence from $0^{\circ }$ to $60^{\circ }$. The structure is found to be stable up to $30^{\circ }$ both for TE and TM modes. The first band of the triple band PC exhibits higher angular stability as compared to the second and third bands (see Figure 15b). The first band shows angular stability of up to $\theta =60^{\circ }$ for both TE and TM modes, but BW is reduced to half with PCR > 90%. This performance of multiband terahertz polarisation converters can be of significant advantage over the single band terahertz converters in multiband applications like MIMO antennas.

## 4. Conclusions

A highly efficient new ultrathin wide-broadband reflective type polarisation converter metasurface consisting of asymmetrically oriented planar double semicircular goblet-shaped resonators. The proposed PC is printed on a PDMS dielectric substrate backed by a gold layer and performs linear polarisation conversion over the frequency range of 0.72–0.99 THz with PCR > 95% and has oblique angle stability up to >$40^{\circ }$ both for TE and TM modes. In fact, according to the authors knowledge this is the only design that shows an ideal polarization conversion (of 100% PCR) over a bandwidth of 95 GHz with angular stability > $45^{\circ }$. Electromagnetic simulations have been presented, followed by a detailed description of the physical mechanism governing the polarisation converter to justify such a large operational BW. In support of the polarisation conversion principle, U-V decomposition analysis and surface current distributions on the top metasurface and bottom ground plane have been carried out to show the impedance imbalance leading to rotation of incident electric field polarization by about $90^{\circ }$. In addition to that, the dual and triple broadband polarisation conversion have been achieved by semi-circular ring addition and slotted geometry. The dualband quad ring structure rotates y- or x-polarized incident EM waves to x- or y-polarized reflected EM waves over the frequency ranges of 0.70–1.08 and 1.61–1.76 THz with PCR > 95% and angular stability > $30^{\circ }$ for both the bands. The slotted DSGR unit cell shows the triple broadband polarization conversion from 0.67–0.85, 1.04–1.11, and 1.62–1.76 THz with PCR > 95% and oblique angle stability up to $30^{\circ }$. The first band of quad ring and slotted DSGR unit cell exhibits extra angular stability of up to >$60^{\circ }$ with PCR > 85%, but bandwidth is reduced to half. A state-of-the-art performance comparison table of the proposed PC with previously reported structures have been given in Table 5 where only simulated results of previous designs are taken into consideration. To claim the fabrication tolerance, parametric analysis has been performed, which reveals that the proposed design is insensitive to slight changes in the physical dimensions. The proposed converter may find potential applications in terahertz spectroscopy and terahertz imaging. This design is simple, easy to modify to implement single and multi broadband polarisation conversion with high PCR at terahertz regime. In the future, the proposed converter will be integrated with terahertz antennas for the analysis of the conformal nature and mutual coupling reduction in terahertz MIMO antennas by applying coding metasurface.

## Figures and Tables

**Figure 1 micromachines-13-01547-f001:**
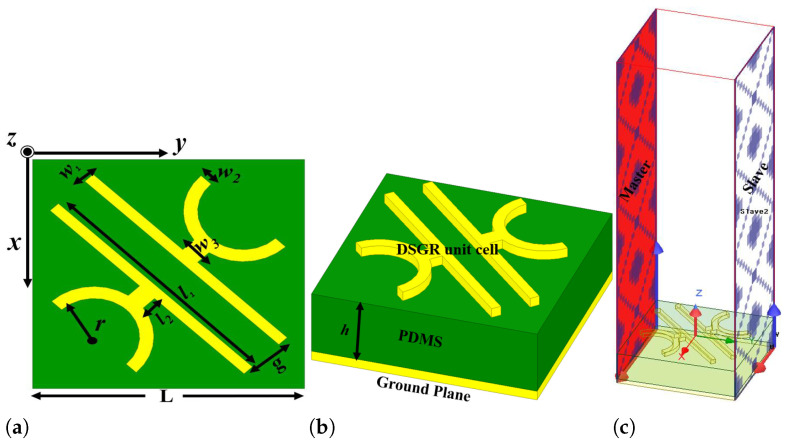
Schematic of the proposed converter (**a**) 3D view (**b**) front view (**c**) HFSS simulation setup of the PC.

**Figure 2 micromachines-13-01547-f002:**
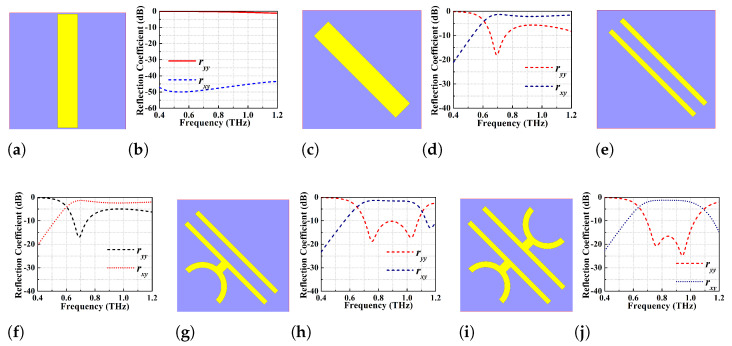
Proposed polarisation converter design steps with corresponding co-pol ($r_{yy}$) and cross−pol ($r_{xy}$) reflection coefficients (**a**,**b**) rectangular gold patch along the x-axis, (**c**,**d**) $45^{\circ }$ rotated rectangular patch w.r.t x-axis, (**e**,**f**) rectangular slot of width 10.5 μm employed to $45^{\circ }$ rotated gold patch, (**g**,**h**) a semi-circular ring of width 5 μm is added to one of the metallic strips, (**i**,**j**) adding two semi-circular rings of same width to individual strips in mirror symmetric configuration.

**Figure 3 micromachines-13-01547-f003:**
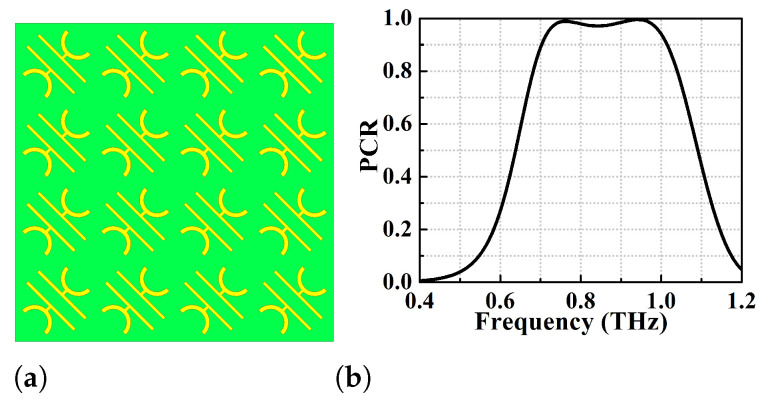
(**a**) Schematic of periodic arrangement of the metasurface unit cell (**b**) polarisation conversion efficiency analysis of the proposed metasurface corresponding to $l_{1}$ = 117 μm.

**Figure 4 micromachines-13-01547-f004:**
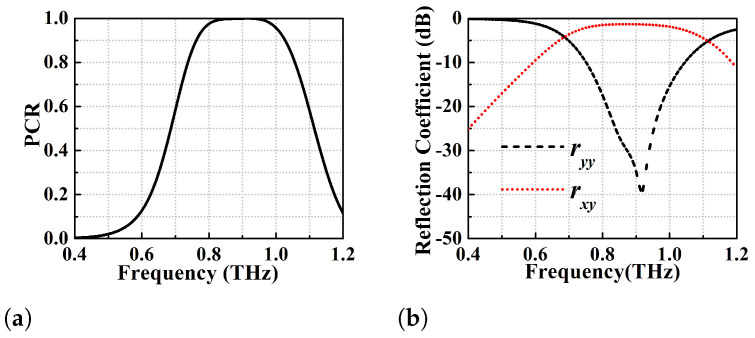
Ideal polarisation conversion efficiency analysis of the optimized design of $l_{1}$ = 110 μm (**a**) 100% PCR (**b**) co-pol $\left(r_{yy}\right)$ and cross-pol $\left(r_{xy}\right)$ reflection coefficients.

**Figure 5 micromachines-13-01547-f005:**
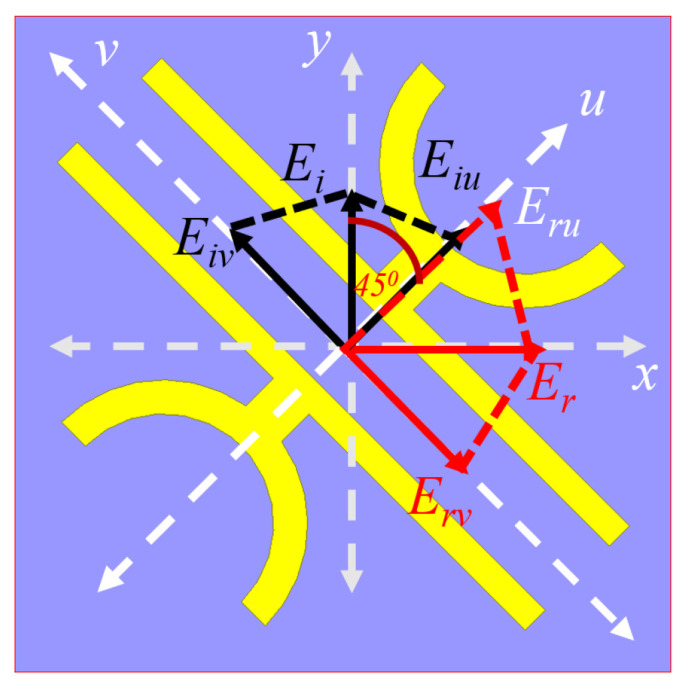
U-V decomposition analysis of the proposed PC.

**Figure 6 micromachines-13-01547-f006:**
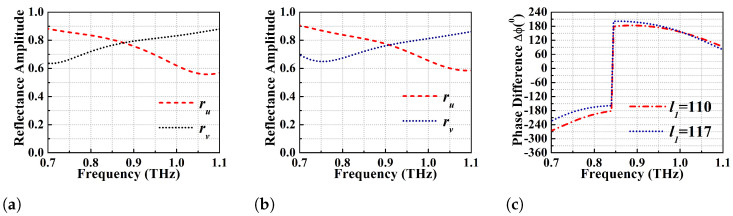
Magnitude of the orthogonal reflection coefficients $r_{u}$ and $r_{v}$ for (**a**) $l_{1}$ = 117 μm, (**b**) $l_{1}$ = 110 μm, (**c**) the phase difference between $r_{u}$ and $r_{v}$ in the PCR defined region.

**Figure 7 micromachines-13-01547-f007:**
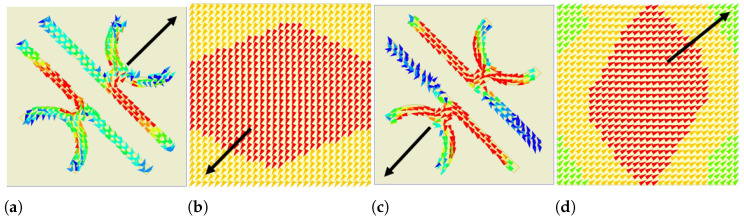
Surface current distribution at 0.755 THz (**a**) top metasurface (**b**) bottom ground plane, surface current at 0.94 THz (**c**) top metasurface (**d**) bottom ground plane.

**Figure 8 micromachines-13-01547-f008:**
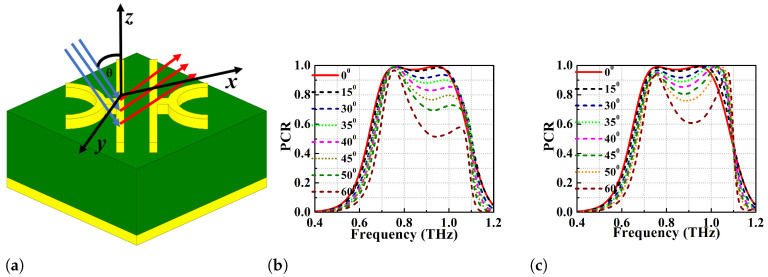
Angular stability performance analysis of the proposed PC (**a**) schematic representation of the metasurface under oblique angle incidence, angular stability at $l_{1}$ = 117 μm (**b**) TE mode (**c**) TM mode.

**Figure 9 micromachines-13-01547-f009:**
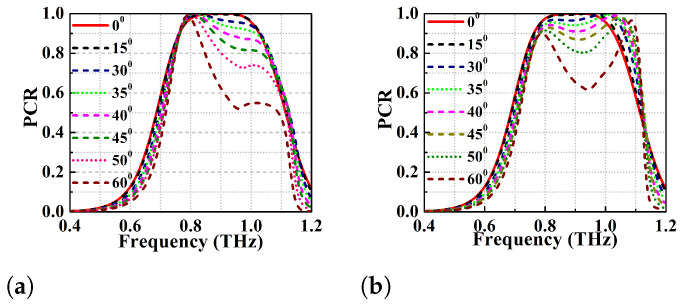
Angular stability at $l_{1}$ = 110 μm (**a**) TE mode (**b**) TM mode.

**Figure 10 micromachines-13-01547-f010:**
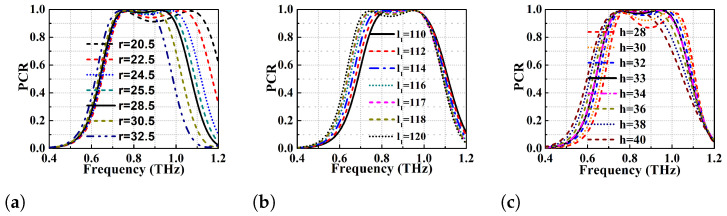
Simulated PCR of the proposed PC at different (**a**) radius of the ring μm (**b**) strip length ($l_{1}$) μm (**c**) height of the dielectric substrate μm.

**Figure 11 micromachines-13-01547-f011:**
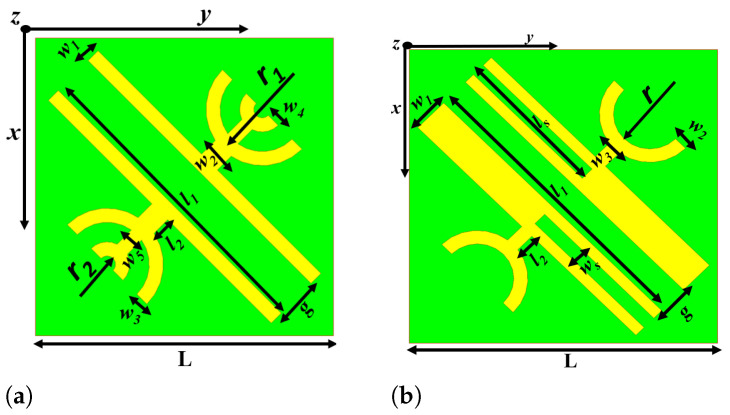
Schematic of multi broadband polarisation converters (**a**) Dual-band PC (**b**) Triple band PC.

**Figure 12 micromachines-13-01547-f012:**
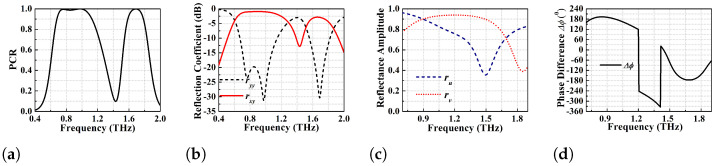
Proposed dual-band polarisation conversion performance analysis (**a**) PCR (**b**) $r_{yy}$ and $r_{xy}$ (**c**) $r_{u}$ and $r_{v}$ (**d**) phase difference between $r_{u}$ and $r_{v}$.

**Figure 13 micromachines-13-01547-f013:**
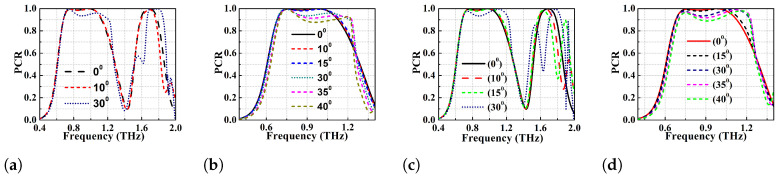
Oblique angle stability analysis of the dual band PC (**a**) TE mode (**b**) TE mode for 1st band (**c**) TM mode (**d**) TM mode for 1st band.

**Figure 14 micromachines-13-01547-f014:**
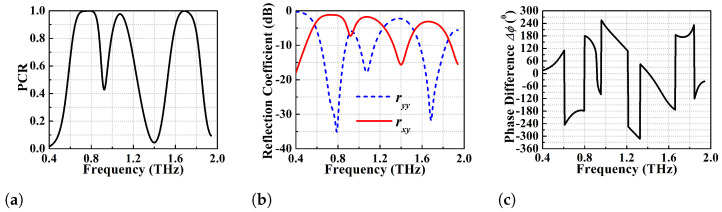
Performance analysis of the triple band PC (**a**) PCR (**b**) co-pol $\left(r_{yy}\right)$ and cross-pol $\left(r_{xy}\right)$ reflection coefficients (**c**) phase difference between $r_{u}$ and $r_{v}$.

**Figure 15 micromachines-13-01547-f015:**
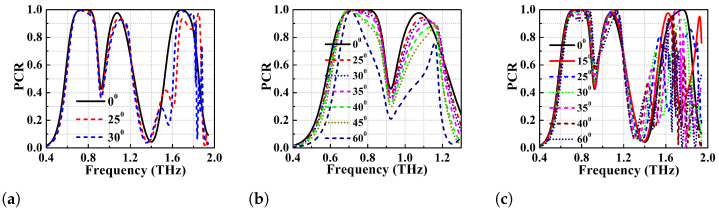
Angular stability analysis of the triple band PC (**a**) PCR for TE mode (**b**) PCR for TE mode in 1st and 2nd band (**c**) PCR for TM mode.

**Table 1 micromachines-13-01547-t001:** Design parameters of the proposed broadband polarisation converter.

L (μm)	$w_{1}$ (μm)	$w_{2}$ (μm)	$w_{3}$ (μm)	t (nm)	$l_{1}$ (μm)	$l_{2}$ (μm)	r (μm)	g (μm)	h (μm)
116	5	9	6	200	117	6.5	25.5	10.5	33

**Table 2 micromachines-13-01547-t002:** Proposed polarisation converter design steps with corresponding co-pol ($r_{yy}$) and cross-pol ($r_{xy}$) reflection coefficients in the frequency range 0.72–0.99 THz.

Reflection Coefficient	Step I	Step II	Step III	Step IV	Step V
$r_{yy}$ (dB)	0	<−5	≤−5	<−10	<−16
$r_{xy}$ (dB)	<−48	≤−2.5	<−2.5	>−2.5	>−2

**Table 3 micromachines-13-01547-t003:** Designed Parameters of the Proposed Dual Broadband Polarisation Converter.

L (μm)	$w_{1}$ (μm)	$w_{2}$ (μm)	$w_{3}$ (μm)	$w_{4}$ (μm)	$w_{5}$ (μm)	$l_{1}$ (μm)	$l_{2}$ (μm)	$r_{1}$ (μm)	$r_{2}$ (μm)	g (μm)	h (μm)
106	5	8	5	5	5	112	6.5	15	8	13	38

**Table 4 micromachines-13-01547-t004:** Designed Parameters of the Proposed Triple Broadband Polarisation Converter.

L (μm)	$w_{1}$ (μm)	$w_{2}$ (μm)	$w_{3}$ (μm)	$w_{s}$ (μm)	$l_{s}$ (μm)	$l_{1}$ (μm)	$l_{2}$ (μm)	r (μm)	g (μm)	h (μm)	t (nm)
110	13	5	5	6.875	55	110	7	18	15.5	36	200

**Table 5 micromachines-13-01547-t005:** State-of-the-art performance comparison table of the proposed converter with previously reported work.

Ref.	Substrate	Periodicity of the unit Cell (in $\lambda_{0}$)	Response of the Converter	Operating Frequency with% PCR	Angular Stability with Reduced PCR	100% Broadband PCR BW	Type of the Converter	Unit Cell Geometry	No. of Dielectric Layers
[29]	Polymide	0.442	Broadband	2.04–5.33, >90	$50^{\circ }$, NA	No	Reflective	Double-split ring resonator (DSRR)	1
[30]	Silica	0.480	Broadband	2.9–4.3, >90	NA	No	Transmissive	Dual Fabry-perot cavity resonator	1
[31]	COC	0.281	Broadband	0.34–1.04, >80	>$30^{\circ }$, 80%	Single frequency band	Reflective	T-shape	1
[36]	PTFE	0.471	Broadband	1.21–2.83, >93	$45^{\circ }$, NA	No	Reflective	Anchor shaped resonator	1
[37]	BCB	0.280	Broadband	0.23–1.17, >90	NA	No	Transmissive	Split disk	1
[38]	Silicon	0.609	Narrowband	only at 0.83 THz, NA	NA	No	Transmissive	Asymmetric Silicon pairs	1
[39]	Polymide	0.294	Broadband	2.1–3.6, >95	NA	No	Reflective	Open metal ring	1
[40]	Quartz	0.091	Broadband	1.28–2.13, >85	$50^{\circ }$, 80%	No	Reflective	Sinusoidally slotted graphene	1
[41]	VO${}_{2}$	0.478	Broadband	4.95–9.39, >90	NA	No	Reflective	E-shape	1
[42]	PDMS	0.409	Broadband	0.65–1.58, >80	NA	No	Reflective	DSRR	1
[43]	Polymide	0.356	Broadband	2.10–5.03, >90	$30^{\circ }$, 80%	No	Reflective	I-shape	1
[44]	SiO${}_{2}$	0.270	Broadband	0.4–0.95, >88	$40^{\circ }$, NA	No	Reflective	I-shape	1
[45]	VO${}_{2}$, Polymide	0.407	Broadband	0.912–2.146, >90	NA	No	Reflective	Circular split ring resonator	2
[46]	TOPAS	0.454	Broadband	1.02–2.76, >90	NA	No	Reflective	Split ring	1
[47]	GaAs	0.196	Dual broadband	0.987–1.062, 0.442–0.537 >90	1st band: >$15^{\circ }$, 2nd band: <$30^{\circ }$	No	Reflective	Arrow shape	1
This work	PDMS	Single **0.292**	Single/Dual/ Triple Broadband polarisation conversion	Single broadband (i) for $l_{1}=117\;\mu$m **0.72–0.99, >95** (ii) for $l_{1}=110\;\mu$m **0.85–0.945, 100**	Single >$40^{\circ }$ (TE and TM) >80%	Single 95 GHz	Reflective	DSGR	1
		Dual **0.270**		Dual broadband **0.70–1.08, >95** **1.61–1.76, >95**	Dual Overall: $30^{\circ }$ (TE and TM) 92% 1st band: >$40^{\circ }$, >88%	Dual No			
		Triple **0.277**		Triple broadband **0.67–0.85, >95** **1.04–1.11, >95** **1.62–1.76, >95**	Triple Overall $30^{\circ }$, 90% 1st band: $60^{\circ }$, >90% but BW is reduced to half	Triple 1st band: **90 GHz** 2nd band: No 3rd band: **30 GHz**			

## Data Availability

The data supporting the findings of this study can be made available to the genuine readers after contacting the corresponding authors.

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
