# Peer review of "Single/Dual/Triple Broadband Metasurface Based Polarisation Converter with High Angular Stability for Terahertz Applications"

_micromachines, 2022, doi:10.3390/mi13091547_

Round 1
Reviewer 1 Report
Comments on the manuscript
In this manuscript entitled “single/dual/triple broadband metasurface based polarisation converter with high angular stability for terahertz applications,” the authors presented a systematic study on the polarization converters with the broadband operation in the THz range and weak angular dispersion features. Specifically, they designed hybrid metasurfaces consisting of a metal nanostructures layer and a metal ground plane, which are sandwiched by a dielectric spacer. The manuscript, in general, is interesting and their theoretical results are incremental to the field of polarization control metasurfaces. Besides, the manuscript is technically sound with well-supported conclusions and assertions. However, the language is a little difficult to understand with some flaws in fluency and accuracy, which degrades the whole level of the manuscript.
Thus, this manuscript needs substantial revision before it can meet the scope of Micromachines. My comments and suggestion to the authors are shown below.
1. The idea of “ multiband polarization control + weak angular dispersion” is not new. Similar contents have been investigated extensively in both experiments and simulations for deep subwavelength nanostructures. Thus what is the key selling point of this work? What are your uniqueness and advantages over previous studies? Spectrum band controllable, fabrication more friendly, or others? The authors need to think seriously about these comments and questions, and briefly highlight the selling point in the abstract since it is closely related to the technical innovations and scientific impact of the manuscript.
2. It seems that the authors are unfamiliar with academic writing.
i. First of all, the introduction section is too long, and it has 3-4 pages with redundant information that are most well-known to the photonic community. This is not professional. Please be concise.
ii. Also, the key point information is not well-summarized. “All the above realisations have one thing in common i.e., wide broadband in nature which is not suitable for multifunctional applications.” This sentence does not comprehensively conclude the current challenges.
iii. In addition, the reference numbers are not correct. For example, “Shaoije Ma et al. [37] have demonstrated highly efficient converters with high angular stability.” However, Ref. 37 refers to another work according to the reference list. “37. Grady, N., Heyes, J., Chowdhury, D., Zeng, Y., Reiten, M., Azad, A., Taylor, A., Dalvit, D. & Chen, H. Terahertz metamaterials for linear polarization conversion and anomalous refraction. Science. 340, 1304-1307 (2013)”. Similar mistakes occur throughout the whole manuscript. Thus, please be careful.
3. “radar cross section (RCS), fiber optics, liquid crystal display, nanophotonic sensing”. The term “Radar cross section” only appears one time in this manuscript. No need to include the abbreviation.
4. The name of the simulation software?
5. “Metamaterials are artificially engineered materials that can easily alter the polarisation states of EM waves due to their unprecedented electromagnetic characteristics like negative refractive index and surface wave manipulation by portable planar structures called metasurfaces (2D equivalent of 3D MM) [16–18].” What is MM? Please define the abbreviation first before you use it. Also, some related works on metasurfaces with polarization control are missing ["Full-stokes polarization perfect absorption with diatomic metasurfaces." Nano Letters 21.2 (2021): 1090-1095].
6. “Figure 4. ryy and rxy after ring addition (a) a semi-circular ring of width 5 µm is added to one side of the rectangular patch (b) adding two semi-circular rings of same width to individual strips in mirror symmetric configuration” This sentence is very difficult to understand. What did you mean “after ring addition (a) a semi-circular ring of width 5 µm is added to one side of the rectangular patch…” This expression is questionable.
7. “The reflective type triple band linear co- to cross-polarization converter simulation results have been discussed here and shown in Fig. 18. Fig. 18(a) exhibits that the triple band PC achieves orthogonal polarization conversion with PCR > 95 % for frequency ranges 0.67 − 0.85 (180 GHz BW), 1.04 − 1.11 (70 GHz BW) and 1.62 − 1.76 (140 GHz BW) THz.” What are the advantages of triple band PCs compared with single band PCs?
Reviewer 2 Report
In this manuscript, the authors find a way to build Polarization Converter. The simulation is dependable. It would be better if the authors could do experiments.
Reviewer 3 Report
In this paper, the authors reported the design and characteristics of a novel compact linear super surface polarization converter for terahertz applications. The broadband metasurface consists of an asymmetrically oriented planar double semicircular cup resonator (dsgr) printed on a conductor supported polydimethylsiloxane dielectric substrate. The broadband metasurface can realize single wideband polarization conversion, while the modified and optimized four ring and slot dsgr can realize dual wideband and three wideband orthogonal polarization conversion. I believe that publication of the manuscript may be considered only after the following issues have been resolved.
1. The length of this work is too long and it is not particularly important. It is suggested that the author place them in the attachment.
2. Fig. 2 is the structure of the design. It is suggested that the author directly place the corresponding diagram in the corresponding spectral diagram.
3. It is suggested to draw Fig. 11 and FIG. 12 as a parameter scanning diagram, which can more vividly describe the change law.
4. The introduction can be improved. Some works on metamaterials and theirs related properties should be added such as Phys. Chem. Chem. Phys., 2022, 24, 8846 – 8853; Plasmonics 2015, 10, 1537–1543; RSC Adv., 2022, 12(13), 7821-7829; Plasmonics 2018, 13, 345–352. Appl. Phys. Express 2019, 12, 052015.
Round 2
Reviewer 1 Report
The authors have addressed my comments and improved the manuscript substantially. The manuscript is scientifically sound. Thus, I have no further questions but to give my proposal of acceptance. Good luck to the authors.
Reviewer 3 Report
Accept in present form